# Biofumigation for Fighting Replant Disease-A Review

**Franziska S. Hanschen** [1,*] **and Traud Winkelmann** [2]

1    Plant Quality and Food Security, Leibniz Institute of Vegetable and Ornamental Crops (IGZ), Theodor-Echtermeyer-Weg 1, 14979 Grossbeeren, Germany

2    Institute of Horticultural Production Systems, Section Woody Plant and Propagation Physiology, Leibniz Universität Hannover, Herrenhäuser Str. 2, D-30419 Hannover, Germany; traud.winkelmann@zier.uni-hannover.de

*    Correspondence: hanschen@igzev.de

**Abstract:** Replant disease is a soil (micro-) biome-based, harmfully-disturbed physiological and morphological reaction of plants to replanting similar cultures on the same sites by demonstrating growth retardation and leading to economic losses especially in Rosaceae plant production. Commonly, replant disease is overcome by soil fumigation with toxic chemicals. With chemical soil fumigation being restricted in many countries, other strategies are needed. Biofumigation, which is characterized by the incorporation of Brassicaceae plant materials into soil, is a promising method. We review the potential of biofumigation in the fight against replant disease. Biofumigation using optimized Brassicaceae seed meal compositions in combination with replant disease tolerant plant genotypes shows promising results, but the efficacy is still soil and site-dependent. Therefore, future studies should address the optimal timing as well as amount and type of incorporated plant material and environmental conditions during incubation in dependence of the soil physical and chemical characteristics.

**Keywords:** Brassicaceae; glucosinolates; isothiocyanate; microbiome; Rosaceae; replant problems; soil-borne pathogens

## 1. The Replant Disease Syndrome

After replanting similar crop species at the same site, severe plant growth depression can be observed. This phenomenon has been termed as "replant problems" (then including soil structural and chemical problems [1]), "soil sickness", "soil decline", "soil fatigue", or "replant disease". Since partial or full soil disinfection can restore plant growth in most cases, biological living agents are the most likely cause of replant problems. According to Winkelmann et al. [2], replant disease is "the harmfully disturbed physiological and morphological reaction of plants to soils that faced alterations in their (micro-) biome due to the previous cultures of the same or related species." This definition implies the previous culture as the starting point of replant disease, which affects the soil microbial and mesofauna communities by root exudates or rhizodeposits and decomposition of plant parts. Thus, replant disease can be classified as negative plant-soil feedback [3] and is mainly due to microbial dysbiosis [2].

Replant disease has been reported for many horticultural and forestry crops, being especially pronounced in fruit orchards with apple [2,4], peach [5,6] and cherry [7,8], but also affecting roses [9,10], grapevine [11,12], asparagus [13], medicinal plants like *Rehmannia glutinosa* [14], and several forestry tree species [15].

The etiology and definite causes of replant disease are still not fully understood. It is considered as a disease complex and is strongly influenced by the plant species and genotype

as well as by soil properties including soil texture, pH, organic matter content, and aeration or water saturation [2]. The plant as the initiator of replant disease suggests that autotoxicity is involved. This is caused by the release of chemicals, often phenolic secondary metabolites, which are toxic to the same and related plant species [3,16]. These autotoxins were shown to be rapidly degraded by rhizosphere and soil microbes [17], resulting in shifts in microbial community composition. In consequence, the accumulation of pathogenic microorganisms as well as the absence of beneficial, plant growth-promoting microorganisms have been reported as associations of replant diseases of several plant species (reviewed by [2,15]). Frequently mentioned pathogenic fungi and oomycetes include species of the genera *Pythium*, *Fusarium, Ilyonectria* (and other *Cylindrocarpon*-like fungi), and *Rhizoctonia* (e.g., [13,18–20]). Bacterial genera that have been associated with replant situations comprise amongst others, *Bacillus* and *Pseudomonas* [3]. With the arrival of new sequencing technologies, replanted soils have recently been subjected to concise microbial community analyses revealing pronounced changes in their structure as well as functions (e.g., [21–24]). Nematodes can contribute to apple replant disease either as phytopathogenic nematodes or free-living nematodes shaping microbial communities [25,26]. Studies on other soil organisms, especially of the mesofauna, are needed to better understand the complex changes in soil biota in replant situations.

The most obvious counteraction against replant disease is crop rotation or changing of sites. This is, however, no longer possible in many cases, especially in central production areas for fruit or wine in which large investments are taken to set up modern orchards with irrigation systems and nets for protection against hail, for example. Intercropping can help to mitigate replant disease by repelling nematodes or by increasing the diversity of soil biota [15,27]. Similarly, the biodiversity in the soil can be increased by soil amendments, typically with compost [28–30]. Anaerobic soil disinfection, i.e., the incorporation of organic carbon under water saturation and sealing with plastic foils that leads to oxygen depletion by facultative anaerobes, was found to be an effective countermeasure against replant disease for instance in apple and cherry [31,32]. Another strategy would be breeding for replant disease tolerance [33,34], but this is time-consuming and difficult as long as the causes and etiology are not resolved. Soil disinfection by heat or chemical means is effective, but ecologically harmful and expensive. Chemicals used for soil fumigation are toxic and nowadays include mainly dazomet or metam sodium (both releasing methyl isothiocyanate) as well as 1,3-dichloropropene/chloropicrine [23,35–37]. Interest is shown in developing sustainable management options to potentially replace chemical soil fumigation. Biofumigation that is based on the release of toxic metabolites from biological material of members of the Brassicaceae plant family, is one of the management options for mitigating replant disease.

## 2. Biofumigation

Upon biofumigation practice, fresh glucosinolate-rich Brassicaceae crops, are chopped and incorporated into the soil in order to achieve natural isothiocyanate formation. Alternatively, Brassicaceae seed meals can be applied [38]. Typical biofumigation crops are mustards such as *Brassica juncea, Sinapis alba, Eruca sativa* or *Raphanus sativus* varieties [39,40]. Indian mustard (*Brassica juncea*), which is rich in allyl glucosinolate, a precursor to allyl isothiocyanate, was most effective in bioassay screenings of Brassicaceae cultivars [41–43].

The term "biofumigation" was coined by J. A. Kirkegaard in 1993 [44]. In the mid-nineties of the 20th century, the first studies on biofumigation were performed where some antiherbicidal potential by formation of volatile glucosinolate hydrolysis products in soil was observed [45]. By incorporating Brassicaceae plants into the soil, isothiocyanates and other compounds are released by enzymatic hydrolysis from glucosinolates, secondary plant metabolites occurring in Brassicales plants [46]. So far, around 137 glucosinolates have been identified and according to their variable side chain they can be classified into aliphatic, aryl(aliphatic), and indole glucosinolates [47]. In Brassicaceae plants, glucosinolates are present in all plants parts but their profile and levels differ enormously within plant organs, ontogenetic stages, species, and varieties [48–50]. For example, ripe seeds of Indian mustard

had a glucosinolate content of 61 μmol/g dry weight (DW), while at flowering stage, mustard stems, roots, and leaves had only around 5, 5, and 4 μmol/g DW, respectively. However, the contents in leaves and roots increased to the "green seeds in pods" stage to approximately 14 and 8 μmol/g DW [48].

The hydrolysis of glucosinolates is initiated when glucosinolates come into contact with myrosinase upon tissue disruption. This β-D-thioglucosidase cleaves β-D-glucose and the intermediary-formed aglucon spontaneously degrades to isothiocyanate or nitrile. Depending on the glucosinolate structure and the presence of specifier proteins, epithionitriles or thiocyanates can also be released [51,52] (Figure 1a). In plants with no or low specifier proteins activity, isothiocyanates usually are the main products [53]. Thiocyanate ions (SCN⁻, demonstrating weed suppressive effects) can also be released from instable isothiocyanates such as 4-hydroxybenzyl isothiocyanate (released from 4-hydroxybenzyl glucosinolate in *Sinapis alba*) [54,55] (Figure 1b). Isothiocyanates on the other hand have antimicrobial [56–58], antifungal [58,59], and antinematicidal [40,60] properties.

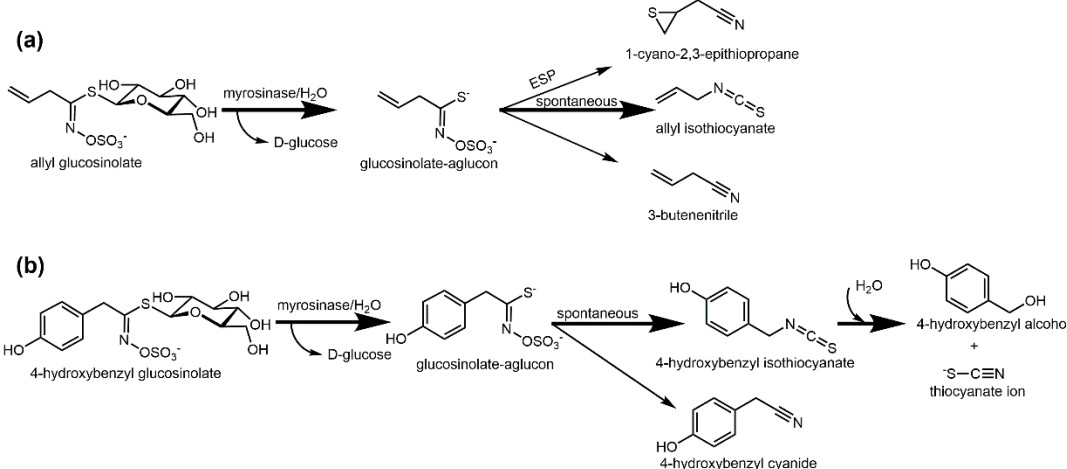

**Figure 1.** Enzymatic hydrolysis of glucosinolates during biofumigation. (**a**) Hydrolysis of allyl glucosinolate from *Brassica juncea* to the corresponding isothiocyanate, nitrile, or, if epithiospecifier proteins (ESP) are present, to the corresponding epithionitrile; (**b**) Hydrolysis of 4-hydroxybenzyl glucosinolate from *Sinapis alba* and hydrolysis of the released 4-hydroxybenzyl isothiocyanate to the corresponding alcohol and thiocyanate ions.

Several factors affect the isothiocyanate formation in the soils: Next to the initial glucosinolate concentration of the plant material, which is usually highest before flowering, the amount of plant material, the myrosinase activity of both plant material and soil, the extent of tissue disruption, the soil temperature, and the water content affect the hydrolysis [38,50,61,62]. Therefore, isothiocyanate levels in soils after biofumigation can range widely from 1 to 100 nmol isothiocyanate/g soil [38]. Calculated effective values for soil sterilization with methyl isothiocyanate range from 517 to 1294 nmol/g soil [63]. For *Verticillium dahliae* control, a necessary allyl isothiocyanate concentration of 150 nmol/g soil was estimated [42]. Therefore, for soil disinfection via biofumigation, high isothiocyanate levels are needed. Biofumigation with Brassicaceae green manure in this respect is often not efficient in reaching adequately high isothiocyanate levels in the soil, as the isothiocyanate-release efficiency of Brassicaceae biomass is typical below 5% due to insufficient cell disruption [38,62,64]. Moreover, the conversion rate can vary between glucosinolates, cover crops and years of cultivation [65]. Thus, the crucial factor is the release of isothiocyanates into the soil and not the glucosinolate levels of the plants themselves [62]. By optimizing the preparation of soil and tissue disruption, higher conversion rates can be achieved. For example, a total isothiocyanate concentration of 91 nmol/g field soil was reached by irrigating the soil 2 days prior to biofumigation (30 mm), then grinding the above-ground high-glucosinolate *Brassica juncea* tissue (by using a rotating flail mulcher running at high revolutions and low ground speed and grinding the plant material to a maximal size of 3 × 3 cm), followed by immediately incorporating

it into the first 10 cm with a rotary hoe and consolidating the soil with two passes of a rubber-tyre roller. Finally, the soil again was irrigated (18 mm) within 3 h [44,62,66]. Nevertheless, often a more practical approach is the use of seed meal to overcome these obstacles: Brassicaceae seeds have higher levels of glucosinolates compared to fresh plant materials [48]. Thus, in Brassicaceae seed meals optimized for biofumigation glucosinolates range from 170 μmol up to 303 μmol/g seed meal [67,68].

Due to the tissue already being homogenized, the hydrolysis of glucosinolates from seed meals after water addition is more efficient than in fresh material. More finely ground *B. juncea* cv. Pacific Gold seed meal (≤ 1mm) released allyl isothiocyanates at a higher rate compared to coarse seed meal (2–4 mm) (5–7 nmol/g soil compared to 4–5 nmol/g soil). Nevertheless the conversion rate detected in that study was still low (303 μmol glucosinolates/g seed meal, 3 mg seed meal/g soil applied = 909 nmol glucosinolates/g soil applied; = conversion rate of 7.7%). However, it has to be kept in mind that in that study, only the head space above the treated soil was sampled and not the soil itself [69]. Further, waterlogging to enhance isothiocyanate release [62] and soil tarping with plastic foils to keep the volatile compounds in the soil [67] are recommended. Usually, the release of glucosinolate hydrolysis products is fast and in many studies the highest isothiocyanate levels were detected in the first few hours after the biofumigation treatment [53,62,63,65,70,71]. Release of thiocyanate ions ($SCN^-$) is slower, and $SCN^-$ is more persistent in the soil [63].

Not only their formation but also isothiocyanate degradation has to be considered for successful results. Especially lipophilic isothiocyanate levels can decline due to sorption to soil particles and soils rich in organic matter absorb more isothiocyanates [38,72–74]. The sorption to soil organic matter also seems to influence the vapor concentration of isothiocyanates in soil (or in the headspace) [75,76], which also reduced the disinfestation efficacy of isothiocyanates in volatile toxicity assays where only isothiocyanates in the gaseous phase contacted the test organism [77]. Therefore, sandy soils with usually low organic matter content will reach higher isothiocyanate peak concentrations compared to soils with high organic matter (for example peat). The disinfestation efficacy of isothiocyanates also depends on the temperature: Using volatile toxicity assays, isothiocyanates were more toxic at higher temperatures compared to lower temperatures (5–20 °C tested, in vitro and with soil [moisture content 75% of field capacity; sand (pH 7.2, 3.26% organic matter), loam (pH 4.9, 6.77% organic matter), and peat (pH 5.69, 31.55% organic matter) tested]) [77]. Again, due to lower organic matter content, assays in sandy soils were more effectively compared to peat soils [77]. Moreover, some of the non-sorbed isothiocyanates may escape due to evaporation [38], but probably most isothiocyanates are degraded due to biodegradation [73,78,79] with chemical degradation playing a minor role [53]. Repeated treatment of soils with Brassicaceae crops (or the same compounds) can stimulate biodegradation [44,79]. Biodegradation increases with elevated pH combined with elevated calcium levels in the soils [44]. Thus, for successful biofumigation, optimal plant material, pretreatment; dosage; weather; and also the soils and their preparation, determine the outcome.

However, several studies could not directly correlate the effects of Brassicaceae biofumigation with glucosinolate or isothiocyanate contents in the treated soils [80–83]. These studies imply that shifts in the microbial community structure are responsible for the effects of biofumigation resulting in disease suppression [80–82,84]. In addition, *Brassica* green manure crops effectively incorporated soil mineral nitrogen that may otherwise leach to the groundwater. Thus, when later incorporated into the soil, *Brassica* materials can provide a source of organic nitrogen [85].

Other compounds formed during Brassicaceae biomass decomposition may also contribute to biofumigation effects. Brassicaceae plants were shown also to release other volatile sulfur-containing compounds. In addition to methanthiol, carbondisulfide, dimethylsulfide and dimethyldisulfide were generated after soil incorporation of Brassicaceae plants [86–88]. Toxic effects have been shown for these compounds on soil microorganisms [89,90]. Dimethyldisulfide is also the active component of an approved chemical fumigant in the USA [91]. Probably, these volatile compounds are degradation products of sulfur containing amino acids such as *S*-methyl-ʟ-cysteine sulfoxide inherently formed in Brassicaceae plants [92,93].

One important aspect for the implementation and economic consideration of biofumigation for the agricultural and horticultural practice is the availability of the plant material that needs to be incorporated into the soil. If Brassicaceaes are grown as a rotation crop, farmers lose time for their cash crops. Thus, and also due to the fact that seed meal or oil-less seed cakes contain high amounts of glucosinolates, the use of these by-products of the biofuel production can enable the provisioning of the required biomass [94,95].

## 3. Effects of Biofumigation on the Soil Biota

Biofumigation treatments affect organisms in the soil. Next to intended effects on plant pathogens, beneficial soil invertebrates can also be affected. 2-Phenylethyl isothiocyanate showed acute toxicity on the soil arthropods *Folsomia candida* and *Protaphorura fimata* [96], the isopod *Porcellio scaber* [97], and the earthworm *Eisenia andrei* [98,99].

One of the most investigated effects of biofumigation is the nematicidal effect, which was recently reviewed by Dutta et al. [100]. Especially Brassicaceae plants releasing the aliphatic allyl isothiocyanate as well as aromatic isothiocyanates such as 2-phenylethyl and benzyl isothiocyanate were promising against these nematodes, but not all stages of the pest are equally susceptible to the treatment [100]. Dutta et al. concluded that biofumigation with Brassicaceae tissues is helpful in plant parasitic nematode control, but that it is unlikely that biofumigation alone will eliminate plant parasitic nematodes in soil. However, in combination with other techniques such as soil solarization or minimal use of nematicides, biofumigation may enable acceptable plant parasitic nematode control [100].

The soil microbial community is important for plant health. Both pathogenic and beneficial strains affect plant growth and health [101]. A shift in microbial community composition with the accumulation of pathogenic microorganisms and the absence of plant growth-promoting microorganisms are linked with replant diseases of several plant species [2,15]. Biofumigation can alter the soil microbial community. Isothiocyanates were reported to inhibit nitrifying bacteria in in vitro bioassays at a dose of 10 μg isothiocyanate/g soil (= 101 nmol allyl isothiocyanate/g soil) depending on the soil type (using sandy- and clay-loam soil, pH 5.9 and pH 7.5, respectively; soils moistened to −480 kPa, incubation at 15 °C for up to 42 days) [102]. Interestingly, in laboratory experiments, 0.32 μmol allyl glucosinolate/g soil [slightly loamy sand and sandy soil (pH 4.8–5.3), water holding capacity 100%] affected the soil microbial communities even stronger than in combination with myrosinase (0.16 μmol/g soil + 0.02 units of myrosinase/g soil), which released the isothiocyanate (room temperature, sampling after 7 days) [53]. Moreover, Siebers et al. reported a decline in soil microbial diversity as accessed by next generation sequencing (sampling after 7–28 days) in a laboratory experiment after soil (loamy sand, pH 6.1) treatment with a rapeseed extract (RSE) rich in glucosinolate hydrolysis products (33 μL RSE/g soil (incubation at 21 °C, moisture less than 18%, RSE addition every 3 days for up to 28 days; in sum ~575 nmol goitrin and ~366 nmol sinapic acid choline ester/g soil added). However, when cultivating surviving fungi and bacteria from treated soils, many of these strains could mobilize phosphate from insoluble sources and had growth-promoting properties on *Arabidopsis thaliana* [103]. Therefore, one important role of glucosinolate hydrolysis products in the efficiency of biofumigation seems to be the potential to favor beneficial microbiota. While metham sodium treatment reduced soil microbial activity in pot experiments (300 μg/g sandy loam soil, pH 7.2, water holding capacity set to 45%, sampling after 3, 15, and 60 days at 23 °C), an increase in soil microbial activity and specific changes in ascomycetes strain abundance were reported after biofumigation with broccoli leaves in a laboratory experiment (15 mg homogenized broccoli leaves/g dry soil, water holding capacity set to 45%) [82]. This effect was probably due to microbial responses to C-substrates, as the response to myrosinase treated broccoli was less pronounced [82]. Organic amendments such as (defatted) seed meals add organic carbon and nitrogen into the soil that are easily available for soil microbial degradation [94]. Moreover, biofumigation with rapeseed meal increased soil content of $NO_3^-$, available P and available K [104]. Thus, increased soil respiration rates as well as enzymatic activities (for example β-glucosidase) were observed in the first month after biofumigation with

*Brassica carinata* seed meal or sunflower seed meals, both obtained from a biofuel byproduct (3 t/ha applied on clay soil, tillage of soil) [105]. Four weeks after biofumigation in field experiments using Indian mustard and radish, there was a shift in soil bacterial community and even more so in fungal community composition: some strains vanished while other strains were promoted due to biofumigation (sandy soil and sandy loamy sand, biofumigation at full flowering of cover crops) [71]. In another field experiment, biofumigation with mustard (3.5 kg/m² of cut material) increased the biodiversity in bacteria and fungi compared to control and fumigated soils, as observed by denaturing gradient gel electrophoresis (DGGE) [106]. Here, treatment with mustard having a glucosinolate content of 38.5 μmol/g DW (being mainly 3-butenyl glucosinolate) was similarly effective in the control of *Fusarium oxysporum* compared to soil fumigation with hymexazol [106]. Biofumigation with rapeseed meal reduced disease incidence of *Phytophthora* blight and significantly increased yield in pepper in a field experiment (loam clay soil, pH 7.2, 0.4% w/w of rapeseed meal incorporated, irrigated after incorporation, covered with plastic foil), although no reductions in *Phytophthora capsici* counts were observed. However, the biofumigation increased richness and bacterial diversity, while it decreased fungal diversity. Thus, changes in soil microbial community structure were hypothesized to be responsible for the disease suppression. The group further reported a negative correlation between soil bacterial diversity and disease incidence of Phytophthora blight [107]. In this experiment, biofumigation of soil pots with rapeseed meal (soil pH 7.2, 4 g rapeseed meal/kg dry soil; water 50% of water holding capacity, soil covered with plastic film after incorporation, incubation at 25 °C for 20 days) increased soil bacterial diversity, bacterial populations including *Bacillus* and *Actinobacteria*, and reduced *Phytophthora capsici* and disease incidence [107]. The use of integrated biofumigation with an antagonistic strain (*Bacillus amyloliquefaciens*) (application of strain after biofumigation) further increased disease suppression effectiveness of biofumigation [107]. Repeated biofumigation with *B. carinata* pellets (Biofence®) and *Sinapis alba* green manure (clay loam, pH 6.4, treatments over three growth periods) showed the highest increase in total bacteria, actinomycetes and *Pseudomonas* ssp. in treated soils compared to soils treated with other non-*Brassica*-based organic amendments [108]. Further, *Pseudomonas* ssp abundance was negatively correlated with the growth of the plant pathogen *Rhizoctonia solani* [108]. Mowlick et al. suggested that *Clostridia*, members of the *Firmicutes*, play an important role in the control of spinach wilt. *Clostridia*-induced organic acid release was discussed as a possible mode of action to explain the effects of biofumigation (*B. juncea*) and *Avena sativa* green manure treatment [109].

Several of the bacterial genera that were observed to be favored due to biofumigation, such as *Pseudomonas*, are known to have beneficial properties. *Pseudomonas* spp. are beneficial bacteria for plant growth as they act as antagonists against soil pathogenic fungi and enhance sulfate uptake [30,110,111]. Moreover, several members of the phylum *Actinobacteria* with plant growth-promoting properties are involved in soil-borne disease suppression [30,112]. Therefore, biofumigation-induced increase in plant growth-promoting and disease-suppressing bacteria seems to be an important mechanism in biofumigation efficiency.

## 4. Efficacy of Biofumigation on Replant Disease

Measuring the effectiveness of a management strategy against replant disease is not a trivial task. Up to now, a comparison of plant growth in replant soil and disinfected replant soil seems to be the most reliable measure (e.g. [23]). Early diagnostic tools were suggested using microscopic preparations of apple roots [113], however, they might not allow to quantify the severity of the disease. A possible approach is to develop early genetic markers on the transcriptional level.

Table 1 summarizes the literature investigating biofumigation for fighting replant diseases of different plant species. Due to the worldwide economic relevance, most studies published so far addressed apple replant disease. Generally, it has been shown that plant growth and fruit yields were significantly improved by biofumigation treatments, especially if Brassicaceae seed meal was used. Nevertheless, due to these treatments, considerable amounts of organic material are added to the

soils, which also may contribute to the positive treatment effects due to improved soil structure and provision of nutrients [44]. For example, soil aggregate stability and water infiltration in sandy soils were described to be improved after biofumigation [44].

**Table 1.** Overview of studies using biofumigation as counteraction of replant disease. Ref.-Reference.

| Kind of Replant Disease | Bio-Fumigation Treatment | Environmental Conditions | Measurement of Efficacy by | Efficacy | Observations | Ref. |
|---|---|---|---|---|---|---|
| Apple replant disease | *Brassica napus* as cover crop | No information provided | Field trials, counts of *Pratylenchus penetrans* and recovery of *Pythium* from soil | No positive effects | No reduction, but rather an increase in *Pratylenchus penetrans* and *Pythium* abundance | [114] |
| Apple replant disease | *B. napus* seed meal 0.1–2.0% | Incubation in the greenhouse (20 °C), no information on soil moisture etc. | Greenhouse pot trials | Increased plant growth, but toxic effects at high concentration. | No consistent reduction in *Pythium* infections, suppression of *Rhizoctonia* and *Pratylenchus penetrans* at 0.1% and increased abundance of fluorescent *Pseudomonas* spp. at 0.1 and 1.0% | [81] |
| Apple replant disease | *B. napus* seed meal 8.5 t ha$^{-1}$ and green manure (for one-three years) | Seed meal incorporation in May 2001, some variants covered by plastic foil (no information on soil temperature/moisture) | Field trial with tree growth and yield measurements | Growth and yield improvement by both, *B. napus* green manure and seed meal treatments, especially when combined with fungicide treatment. | Reduction of ARD associated pathogens, i.e., *Pratylenchus penetrans*, *Pythium*, *Cylindrocarpon*, *Rhizoctonia*, but not *Fusarium* by the combined treatment of *B. napus* seed meal and fungicide, not by green manure. | [115] |
| Apple replant disease | *Brassica juncea* plant material (1–3 years) and *B. napus* seed meal combined with other treatments | No information provided | Field test and greenhouse bio-test of plant growth and yield (field) | Cumulative yield increase in a site-dependent way, mainly by seed meal treatments | Control of *Cylindrocarpon*, *Rhizoctonia* and *Pythium ultimum* by seed meal treatments, best in combination with a fungicide treatment, lower effect on *Pratylenchus penetrans* | [116] |
| Straw-berry replant disease | *B. juncea* cover crop incorpo-rated into the soil | Incorporation of plant residues in April 2002, no further information on soil temperature or moisture | Pot trial and field experiment | Fruit yield as well as vegetative growth parameters increased in the pot and the field trial | *Rhizoctonia* abundance was reduced by mustard treatment, but causes for this kind of replant disease is not clear. | [117] |
| Apple replant disease | Seed meals of *B. juncea*, *Sinapis alba* and *B. napus*; 0.5% (*wt/wt*) | Eight weeks of incubation at 22 ± 3 °C, no information on soil moisture | Greenhouse bio-test in pots | Seed meal improved apple seedling growth, seed meal reduced *Rhizoctonia solani* infection in native but not in pasteurized soil, while *Streptomyces* ssp. increased it | *B. juncea* seed meal was most effective in *Pratylenchus penetrans* suppression and the only seed meal that did not increase *Phytium* populations | [70] |
| Apple replant disease | Seed meals of *B. juncea*, *S. alba* and *B. napus*; 0.5% (vol/vol) | Blending and sieving (< 1 mm) of seed meals, 8 weeks of incubation at 22 ± 3 °C, no information on soil moisture | Greenhouse bio-test in pots | Seed meal-specific effects on *Pythium* and *Pratylenchus penetrans* numbers and infections. | *B. juncea* seed meal suppressed *Pythium* and *P. penetrans* populations. | [118] |
| Apple replant disease | Seed meal of *B. juncea*; 0.3% (*wt/wt*) = 4.5 t ha$^{-1}$ | Fine (<1 mm) and coarse (2–4 mm) seed meal particles incorporated, no further information on soil temperature or moisture | Bio-test in greenhouse, variation of particle sizes of seed meal | Suppression of *Rhizoctonia solani* SG5 (for fine seed meal), *Pratylenchus penetrans* and *Pythium* spp. infections | Biological and chemical effects of the seed meal, increased population densities of *Streptomyces* and more free-living nematodes | [69] |

**Table 1.** *Cont.*

| Kind of Replant Disease | Bio-Fumigation Treatment | Environmental Conditions | Measurement of Efficacy by | Efficacy | Observations | Ref. |
|---|---|---|---|---|---|---|
| Apple replant disease | Seed meals of *B. juncea*, *S. alba* and *B. napus*; 4.5 kg m$^{-1}$ tree row | Incorporation in April 2005, May 2006, April 2007, respectively, tarped with plastic foil for 1 week, no further information on soil temperature or moisture | Field trial with measures of tree diameter and cumulative yield | Significant improvement of tree growth and cumulative fruit yield when seed meals (except for *B. napus*) were combined with fungicide soil drench | Seed meal specific effects, *B. napus* resulted in increased *Pythium* and *Pratylenchus penetrans* densities, whereas *B. juncea* reduced both pathogens as well as *Cylindrocarpon* infections but only when combined with fungicide drench. Without fungicide treatment, *B. napus* and *S. alba* seed meal amendments caused *Pythium* and *B. juncea* caused *Phytophtora* infections. | [68] |
| Apple replant disease | Seed meals of *B. juncea*, *S. alba* and *B. napus*; 0.3% (wt/wt) | Blending and sieving (<1mm) of seed meals, 48 h incubation in plastic bags, no information on soil temperature or moisture | Bio-test in greenhouse | Reduction of apple seedling mortality after *B. juncea* seed meal application in one soil. | Soil-dependent and seed meal-dependent shifts in *Pythium* communities, *S. alba* led to increased *P. ultimum* levels. | [119] |
| Apple replant disease | Seed meal of *B. juncea*; 0.3% (wt/wt) = 4.5 t ha$^{-1}$ | Fine (<1 mm) and coarse (2–4 mm) seed meal particles incorporated, bagged or non-bagged incubation for 48 h, no further information on soil temperature or moisture | Bio-test in greenhouse | Reduction of *Pythium abappressorium* infections, especially in the bagged variants | Suppressiveness of soil was achieved, possibly due to long-term changes in fungal communities, especially promotion of *Trichoderma* spp. | [120] |
| Apple replant disease | Seed meal blends of *B. juncea*, *S. alba* and *B. napus*; 6.7 t ha$^{-1}$ | Incorporation of seed meals once in March 2010 or twice in September 2009 and April 2010, tarped for 1 week, no further information on soil temperature or moisture | Field test of plant growth and yield | Significant increase in tree growth of *B. juncea* + *S. alba*, positive long-term effect (4 years), but mortality if applied few weeks prior to planting. Efficacy superior to chemical fumigation | Effective reduction of *Pratylenchus penetrans* mainly in the first year, *Pythium* infections enduringly reduced. Resilient changes in rhizosphere microbial communities. | [121] |
| Peach replant disease | *B. juncea* plant biomass and canola seed meal cake in a field experiment | Watering before incorporation in June 20, 1 day later tarping, recording of soil temperature during the 2-months treatment (26–34 °C) | Field test of plant growth | Significantly improved tree growth | Better plant health, lower mortality | [122] |
| Apple replant disease | Incorpora-tion of plant material of *B. juncea* and *Raphanus sativus* in the field | Incorporation in May and August 2012 and 2013, no further information on soil temperature or moisture | Greenhouse bio-test of plant growth and field test | Site specific increase in biomass production after biofumigation. | Nutrient effect and stronger shifts in fungal than in bacterial community composition | [71] |
| Apple replant disease | Incorpora-tion of plant material of *B. juncea* and *R. sativus* in the field | Incorporation in May and August 2012 and 2013, no further information on soil temperature or moisture | Field test of plant growth | Site specific effects (only in the tested sandy soil about 150% increase in growth, no significant change in the second soil). | Bacterial genera with increased abundance: *Arthrobacter* (*R. sativus*), *Ferruginibacter* (*B. juncea, R. sativus*). Fungal genera of higher abundance: *Podospora*, *Monographella* and *Mucor* (*B. juncea, R. sativus*) | [27] |
| Apple replant disease | Incorpora-tion of seed meal formulation of *B. juncea* and *S. alba* 1:1 in the field, 2.2, 4.4, 6.6 t ha$^{-1}$ | Incorporation in April 2016, tarping for 2 weeks, soil temperature: 12−14 °C, no information on soil moisture | Field test of plant growth | Significantly improved tree growth, 4.4 t ha$^{-1}$ was optimal | Soil fumigation and seed meal amendments suppressed *Pythium* infection in rootstook-specific way. Long-term effect on soil microbial communities. Beneficial microbes increased due to biofumigation | [67] |
| Apple replant disease | Incorpora-tion of seed meal formulation of *B. juncea* and *S. alba* 1:1, dosage, 2.2, 4.4, 6.6 t ha$^{-1}$ | Incorporation into moist soil (−63 to −92 hPa), incubation in bags for 48 h under greenhouse conditions | Greenhouse bio-test in pots | Significantly improved tree growth at all dosages; no difference between 4.4 and 6.6 t ha$^{-1}$, high efficacy in *P. penetrans* and *Pythium* ssp. control | Geneva rootstocks had less colonization by *Pythium* ssp. or *P. penetrans* compared to Mailling rootstocks; both rootstock genotype and soil treatment affected soil microbiom | [123] |

The efficacy was demonstrated to depend on the soil and site and its prevalent pathogenic organisms. A focus of many studies was on the effects of biofumigation on reducing major causal agents of apple (or strawberry) replant disease, mainly *Pythium*, *Rhizoctonia*, *Cylindrocarpon*, and *Pratylenchus penetrans* [70,115–118,121]. This reduction in several pathogens was attributed to chemical and

nutritional effects of the treatments but also to biological effects, i.e., changes in microbial communities and increased abundance of disease-suppressive microbes, such as fluorescent pseudomonads [81], *Streptomyces* spp. [69] or *Trichoderma* spp. [120]. Sometimes, however, and depending on the amount of amended seed meal and especially on the Brassicaceae plant species and its glucosinolate content and composition, even increased populations of *Pythium* and the nematode *Pratylenchus penetrans* or toxic effects on the cultivated plants have been reported [81]. The latter was probably due to thiocyanate ions, which have phytotoxic activity [54]. In a recent study, apple replant disease incidence declined in soils biofumigated with *Raphanus sativus* or *B. juncea* covering crops for 2 years in a site-dependent manner (field plot experiment, biofumigation twice a year at full bloom into moist soil, mechanical cutting and chopping of plants with a flail mulcher, immediate incorporation with a rotary cultivator; soil layering with rolls of a sowing machine, no soil tarping). In sandy soil (pH 5.2) K, the effect was superior to that of slightly loamy sand soil A with pH 4.8. Biofumigation in slightly loamy sand soil M with pH 5.7 did not increase the growth of indicator plants [71]. This correlated to the shifts in the bacterial and fungal communities (analyzed 30 days after second biofumigation in each year), which were strongest at soil K [71], and thus, again point to the role of the soil microbial community in the cure of replant disease. Later, using amplicon sequencing, bacterial genera (for example *Arthrobacter* or *Ferruginibacter*) and fungal genera (for example *Podospora*) were identified that were increased due to biofumigation with the cover crops and were positively linked with growth of apple M106 plants. In contrast, the bacterial genera *Flavitalea* and the fungal genera unclassified *Pleosporales*, *Cryptococcus*, and *Mucor* were negatively correlated with the growth of M106 plants [27].

Brassicaceae seed meals from a single species so far failed to achieve a similar control of replant disease compared to chemical fumigation (rotovated soil, application of 2.23 kg/m$^2$ of seed meal, coverage with plastic film for 1 week after rotovation) [68]. Appropriately implemented biofumigations with particularly formulated seed meals of different Brassicaceae species performed in soil disinfestation comparably to chemical fumigation treatments [67,68,121]. When comparing seed meal formulations, formulations of *Brassica juncea* [rich in allyl glucosinolate (sinigrin), Figure 1a] in combination with *Sinapis alba* [rich in 4-hydroxybenzyl glucosinolate (sinalbin), Figure 1b] were superiorly compared to *B. juncea-Brassica napus* seed meal [121]. In order to reach effective concentrations of glucosinolate breakdown products, high amounts of Brassicaceae seed meal have to be incorporated into the soils, ranging from 1.5 t ha$^{-1}$ (= 0.1% *wt/wt*) [81] up to 8.5 t ha$^{-1}$ [115], with toxic effects at 30 t ha$^{-1}$ (= 2.0% *wt/wt*) [81]. However, the glucosinolate content is not necessarily linked to the efficacy of the biofumigation on replant disease, and the latest studies observed best effects at reduced seed meal levels of 4.4 t ha$^{-1}$ (seed meal formulation of 1:1 *B. juncea* and *S. alba* with 173 µmol glucosinolates/g, tillage applied on soil (Burch loam, pH 6.8) before and after biofumigation, plastic foil cover after incorporation, temperature 12–14 °C) [67,123]. Part of the observed variability in the efficacy of biofumigation especially in field trials may also be due to other environmental factors influencing plant growth in addition to replant disease, which also differs in severity. From the data presented in Table 1, it becomes obvious that environmental factors, such as soil temperature and moisture were not considered in most of the studies. The important question of tarping in the case of field application and bagging or sealing in the case of laboratory and greenhouse application and its duration was shown to have a dramatic effect on efficacy in terms of pathogen suppressiveness [117]. Moreover, the apple genotype affects the efficacy of disease control, and usage of more tolerant apple rootstocks is recommended [67,123].

## 5. Conclusions

Biofumigation with Brassicaceae seed meal formulations has promising potential in the fight of especially apple replant disease, and interest in this management option may even increase in the future as chemical fumigation in many countries is no longer an alternative. So far, the best results against replant disease have been achieved using a seed meal formulation of *Brassica juncea* and *Sinapis alba* (1:1) at an application rate of 4.4 t/ha in combination with tolerant apple rootstocks. Here, mainly the effect

of biofumigation on the soil microbiome and nematodes is linked to the plant health improvement. However, more research is needed in order to optimize its efficacy, which is also site-dependent. Future studies should address the optimal timing as well as amount and type of incorporated plant material in dependence of the soil physical and chemical characteristics. In-depth studies should unravel the effects of Brassicaceae biomass but also glucosinolates and glucosinolate breakdown products on different key soil organisms in order to come to designed mixtures of plant materials that can be used in effective biofumigation treatments. Concerning the requisition of the high amounts of biomass needed for effective biofumigation treatments, a link to the by-products (seed cakes) of bio-diesel production from Brassicaceae oil crops seems a promising approach. Therefore, breeding of these oilseed crops should consider glucosinolate content and profiles.

**Author Contributions:** Conceptualization, F.S.H. and T.W.; writing—original draft preparation F.S.H. and T.W. All authors have read and agreed to the published version of the manuscript.

**Funding:** F.S.H. is funded by the German Leibniz -Association (Leibniz-Junior Research Group OPTIGLUP; J16/2017).

**Acknowledgments:** Amy Schmiedeskamp is thanked for critically reading the manuscript.

**Conflicts of Interest:** The authors declare no conflict of interest.

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
