# Peer review of "Biofumigation for Fighting Replant Disease- A Review"

_agronomy, doi:10.3390/agronomy10030425_

Round 1

Reviewer 1 Report

The paper is certainly improved after the revision process and I highly appreciate the comphrensive revision of this manuscript.

I recommend the MS under minor revision by suggesting a recent reference that should be quoted at line 103 (after ....compost, references 28,29), line 387 (after .....uptake, references 108,109) and line 388 (after ......soil-borne disease suppression, before full stop).

To speed up access and citability of this manuscript, the Authors and Editors find attached herewith the suggested paper (DOI: 10.1016/j.rhisph.2020.100192).  

Author Response

Reply: Thank you for your feedback! As suggested we incorporated the reference at the proposed positions into the text.

Reviewer 2 Report

The manuscript has been improved from the first version. Yet the previous reviewers’ comments were not sufficiently addressed. Without reviewing diverse environmental factors influencing the efficacy of biofumigation on soil disinfestation and providing conclusive guidelines for efficient biofumigation practices using Brassicaceae seed meals (BSM), the manuscript does not deliver significant values to readers. Specifically in Section 4 of the manuscript, research findings on how BSM formulation, BSM dosage, temperature, soil moisture content, BSM soil incorporation methods, incubation time, and soil tarping affect isothiocyanate concentration profile, soil disinfestation results, and crop yield improvements should be extensively reviewed. Optimal conditions and operations for implementing effective biofumigation using BSM should be given in the conclusion section and in the abstract. Additional editorial comments can be found as follows:

Line 19: Change “resistant” to “resistance”

Line 22: Change “soil sickness” to “soil-borne pathogens”

Line 31: Should be “species.”

Line 42: Change “aeration or water saturation as discussed for apple replant disease [1]” to “and aeration or water saturation [1].”

Line 45: The statement on allelochemicals is not correct. Allelochemicals are physiologically detrimental to other plant species, not to similar or the same species of plants.  They are not autotoxins.

Line 52: Change “associated to” to “associated with.”

Line 78: Change “fresh glucosinolate rich Brassicaceae crops, are crushed” to “glucosinolate-rich Brassicaceae plant materials are chopped”

Line 80: Delete “or dried Brassicaceae plant materials”

Line 81: Delete “for example”

Line 83: Add “a” before “precursor.”

Lines 85-87: Delete the sentence “The term …. observed [44].”

Line 91: Change “Brassicales” to “Brassicaceae”

Line 93: The glycosinolate concentration profile of Indian mustard (Brassica juncea) in different plant parts including the seed meal should be given herein after “…varieties [47-49].”

Line 95: Delete “an”

Line 97: Change “…proteins also epithionitriles or thiocyanates can be released” to “…proteins, epithionitriles and thiocyanates can also be released”

Line 99: Change “From instable isothiocyanates such as 4-hydroxybenzyl isothiocyanate, released from Sinapis alba, also thiocyanate ions (SCN-), which have weed suppressive effects [53,54] can be released” to “Thiocyanate ions (SCN-, demonstrating weed suppression effects) can also be released from instable isothiocyanates such as 4-hydroxybenzyl isothiocyanate [53,54].”

Line 110: Delete “the efficacy of”

Line 114: Change “can range, with 1 to 100 nmol isothiocyanate/g soil, enormously” to “can range widely from 1 to 100 nmol isothiocyanate/g soil.”

Line 115: Change “Calculated values” to “Calculated effective values”

Line 117: Change “calculated” to “estimated”

Line 118: Change “fresh green Brassicaceae manure in this respect” to “Brassicaceae green residues”

Line 119: Change “reaching high isothiocyanate levels in the soil with isothiocyanate-release efficiency being below …” to “reaching adequately high isothiocyanate levels in soil, as the isothiocyanate-release efficiency of Brassicaceae biomass is typically below …”

Line 124: Add “,” after “For example”; Change “could be” to “was”

Line 129: Remove [43,61].

Line 131: Give out the general content of glucosinolates in Brassicaceae seed meal.

Line 133: Change “should be” to “is”

Line 134: Delete “and amount”

Line 136: Change “is” to “was”

Line 139: Change “also use covering of soil” to “soil tarping”

Line 143: Remove “(for example from instable indole isothiocyanates)”

Line 148: Change “the concentration of isothiocyanates above the soils” to “the vapor concentration of isothiocyanates in soil”

Line 149: Change “toxicity” to “disinfestation efficacy”

Line 157: The statement is confusing. Suggest to delete “and sandy soils are more susceptible to biodegradation compared to loamy soil.”

Lines 160-165: Remove this confusing paragraph. Formation of isothiocyante during Brassicaceae residue decomposition has been widely confirmed.

Line 166: Change “Nevertheless, also other compounds should be taken into account to contribute to…” to “Other compounds formed during Brassicaceae biomass decomposition may also contribute to …”

Lines 168-169: Modify the sentence as “In addition to methanthiol, carbondisulfide, dimethylsulfide and dimethyldisulfide were gererated after soil incorporation of Brassicaceae plants [84-86].”

Line 176: Change “intercrops” to “a rotation crop”

Line 180: Delete “(Micro)”

Line 220: Delete “treatments”

Line 233: DW stands for dry weight?  Please define.

Line 233: Delete “nearly”

Line 242: Change “biofumigation with rapeseed meal with in a pot experiment” to “biofumigation of soil pots with rapeseed meal”

Line 251: Change “…ssp abundance negatively correlated with…” to “…ssp. abundance was negatively correlated with the…”

Line 255: Delete “Whether Clostridia might also play a role if no gas-tight plastic cover would be used was not tested”

Line 258: spp.?

Line 280: Change “the biofumigation effects in” to “the effects of biofumigation on”

Line 289: Add the period symbol “.” after “activity [53]”

Line 291: The results were from field plot experiments? Details are needed to clarify the rate of biofumigation using cover crop, soil moisture, temperature, tarping, and treatment time.

Line 304: Change “Among the different Brassicaceae species tested, seed meals formulations, however performed as well as fumigation treatments” to “Appropriately implemented biofumigations with particularly formulated seed meals of different Brassicaceae species performed in soil disinfestation comparably to chemical fumigation treatments”

Author Response

The manuscript has been improved from the first version. Yet the previous reviewers’ comments were not sufficiently addressed. Without reviewing diverse environmental factors influencing the efficacy of biofumigation on soil disinfestation and providing conclusive guidelines for efficient biofumigation practices using Brassicaceae seed meals (BSM), the manuscript does not deliver significant values to readers. Specifically in Section 4 of the manuscript, research findings on how BSM formulation, BSM dosage, temperature, soil moisture content, BSM soil incorporation methods, incubation time, and soil tarping affect isothiocyanate concentration profile, soil disinfestation results, and crop yield improvements should be extensively reviewed. Optimal conditions and operations for implementing effective biofumigation using BSM should be given in the conclusion section and in the abstract.

Reply: Thank you for your feedback and for again investing time in our manuscript! As suggested we emphasized the factors influencing the efficacy of biofumigation in the abstract (lines 23-25 of manuscript with changes highlighted), section 4 (lines 382-389 of manuscript with changes highlighted) and it optimal conditions regarding treatment of replant disease with biofumigation are mentioned in the conclusions (lines 401-404 of manuscript with changes highlighted).

We agree that environmental factors will also affect isothiocyanate release, stability and efficacy of biofumigation. The effect of temperature on the efficacy of Isothiocyanate-mediated disinfestation is now discussed in L 178ff. of manuscript with changes highlighted. The efficacy is also strongly linked to the soil (organic matter), which has been discussed in L 172ff. of manuscript with changes highlighted.

To demonstrate the importance of your point we also added a column to Table 1, in which environmental factors are specifically listed. However, unfortunately data on temperature etc. very often are lacking in studies on the efficacy of biofumigation on replant disease. We hope that with those changes we followed your comment adequately.

Additional editorial comments can be found as follows:

Line 19: Change “resistant” to “resistance”

Reply: Done.

Line 22: Change “soil sickness” to “soil-borne pathogens”

Reply: Done.

Line 31: Should be “species.”

Reply: Corrected

Line 42: Change “aeration or water saturation as discussed for apple replant disease [1]” to “and aeration or water saturation [1].”

Reply: Modified as suggested.

Line 45: The statement on allelochemicals is not correct. Allelochemicals are physiologically detrimental to other plant species, not to similar or the same species of plants.  They are not autotoxins.

Reply: You are right, sorry for this mistake. We have replaced allelochemicals by chemicals. (We wanted to express that the principle mechanisms might be similar as in allelopathy, but your comment was completely correct.)

Line 52: Change “associated to” to “associated with.”

Reply: Done

Line 78: Change “fresh glucosinolate rich Brassicaceae crops, are crushed” to “glucosinolate-rich Brassicaceae plant materials are chopped”

Reply: We exchanged “crushed” for “Chopped but left the rest to stress that only fresh materials are chopped (seeds are ground before application). It now reads “Upon biofumigation practice, fresh glucosinolate rich Brassicaceae crops, are chopped and incorporated into the soil in order to achieve natural isothiocyanate formation.”

Line 80: Delete “or dried Brassicaceae plant materials”

Reply: Done

Line 81: Delete “for example”

Reply: Done

Line 83: Add “a” before “precursor.”

Reply: As, to our knowledge, there is no other precursor to allyl isothiocyanate (no other than allyl glucosinolate), we would like to leave it without “a”.

Lines 85-87: Delete the sentence “The term …. observed [44].”

Reply: We would like to keep this sentence. Sorry, but this time we can’t understand, why it should be removed and would like to keep the historical information.

Line 91: Change “Brassicales” to “Brassicaceae”

Reply: Modified as suggested.

Line 93: The glycosinolate concentration profile of Indian mustard (Brassica juncea) in different plant parts including the seed meal should be given herein after “…varieties [47-49].”

Reply: The information has been added and it now reads: “For example, ripe seeds of Indian mustard had a glucosinolate content of 61 µmol/ g dry weight (DW), while at flowering stage mustard stems, roots, and leaves had only around 5, 5, and 4 µmol/g DW, respectively. However, the contents in leaves and roots increased to the “green seeds in pods” stage to approximately 14 and 8 µmol/g DW [49]

Line 95: Delete “an”

Reply: Deleted.

Line 97: Change “…proteins also epithionitriles or thiocyanates can be released” to “…proteins, epithionitriles and thiocyanates can also be released”

Reply: Changed as suggested.

Line 99: Change “From instable isothiocyanates such as 4-hydroxybenzyl isothiocyanate, released from Sinapis alba, also thiocyanate ions (SCN-), which have weed suppressive effects [53,54] can be released” to “Thiocyanate ions (SCN-, demonstrating weed suppression effects) can also be released from instable isothiocyanates such as 4-hydroxybenzyl isothiocyanate [53,54].”

Reply: Modified to “Thiocyanate ions (SCN-, demonstrating weed suppressive effects) can also be released from instable isothiocyanates such as 4-hydroxybenzyl isothiocyanate (released from 4-hydroxybenzyl glucosinolate in Sinapis alba) [53,54] (Scheme 1b).”

Line 110: Delete “the efficacy of”

Reply: Deleted.

Line 114: Change “can range, with 1 to 100 nmol isothiocyanate/g soil, enormously” to “can range widely from 1 to 100 nmol isothiocyanate/g soil.”

Reply: Changed as suggested.

Line 115: Change “Calculated values” to “Calculated effective values”

Reply: Changed.

Line 117: Change “calculated” to “estimated”

Reply: Changed as suggested.

Line 118: Change “fresh green Brassicaceae manure in this respect” to “Brassicaceae green residues”

Reply: We followed your suggestion of a new order of the wording, but would like to stay with the word “manure” as the crops are purposefully planted, so we think “residues” is not the best choice or wording here. It now reads: Brassicaceae green manure…

Line 119: Change “reaching high isothiocyanate levels in the soil with isothiocyanate-release efficiency being below …” to “reaching adequately high isothiocyanate levels in soil, as the isothiocyanate-release efficiency of Brassicaceae biomass is typically below …”

Reply: Modified as suggested.

Line 124: Add “,” after “For example”; Change “could be” to “was”

Reply: Done

Line 129: Remove [43,61].

Reply: Corrected and deleted.

Line 131: Give out the general content of glucosinolates in Brassicaceae seed meal.

Reply: The sentence “Thus, in Brassicaceae seed meals optimized for biofumigation glucosinolates range from 170 µmol up to 303 µmol/g seed meal [68,69].” was added.

Line 133: Change “should be” to “is”

Reply: Done

Line 134: Delete “and amount”

Reply: Done

Line 136: Change “is” to “was”

Reply: Done

Line 139: Change “also use covering of soil” to “soil tarping”

Reply: Modified as suggested.

Line 143: Remove “(for example from instable indole isothiocyanates)”

Reply: Done

Line 148: Change “the concentration of isothiocyanates above the soils” to “the vapor concentration of isothiocyanates in soil”

Reply: Changed as suggested.

Line 149: Change “toxicity” to “disinfestation efficacy”

Reply: it now reads: … which also reduced the disinfestation efficacy of isothiocyanates in volatile toxicity assays…

Line 157: The statement is confusing. Suggest to delete “and sandy soils are more susceptible to biodegradation compared to loamy soil.”

Reply: Changed as suggested.

Lines 160-165: Remove this confusing paragraph. Formation of isothiocyante during Brassicaceae residue decomposition has been widely confirmed.

Reply: We rephrased this paragraph to reduce confusion. But we would prefer to leave the statement. Indeed, isothiocyanate release occurs during biofumigation. However, the disease suppressing effect does not always correlate with isothiocyanate or glucosinolate levels, so other factors seem to be important, too. Therefore, we would like to keep the paragraph in the revised form. It now reads “However, several studies could not directly correlate the effects of Brassicaceae biofumigation with glucosinolate or isothiocyanate contents in the treated soils [81-84]. These studies imply that shifts in the microbial community structure are responsible for the effects of biofumigation resulting in disease suppression [81-83,85]. In addition, Brassica green manure crops effectively incorporated soil mineral nitrogen that may otherwise leach to the groundwater. Thus, when later incorporated into the soil Brassica materials can provide a source of organic nitrogen [86].

Line 166: Change “Nevertheless, also other compounds should be taken into account to contribute to…” to “Other compounds formed during Brassicaceae biomass decomposition may also contribute to …”

Reply: Done.

Lines 168-169: Modify the sentence as “In addition to methanthiol, carbondisulfide, dimethylsulfide and dimethyldisulfide were gererated after soil incorporation of Brassicaceae plants [84-86].”

Reply: Changed as suggested.

Line 176: Change “intercrops” to “a rotation crop”

Reply: Done.

Line 180: Delete “(Micro)”

Reply: Done.

Line 220: Delete “treatments”

Reply: Done.

Line 233: DW stands for dry weight?  Please define.

Reply: Yes. Is now defined in L95.

Line 233: Delete “nearly”

Reply: Done.

Line 242: Change “biofumigation with rapeseed meal with in a pot experiment” to “biofumigation of soil pots with rapeseed meal”

Reply: Changed.

Line 251: Change “…ssp abundance negatively correlated with…” to “…ssp. abundance was negatively correlated with the…”

Reply: Done.

Line 255: Delete “Whether Clostridia might also play a role if no gas-tight plastic cover would be used was not tested”

Reply: Done.

Line 258: spp.?

Reply: Thank you, we corrected it to spp.

Line 280: Change “the biofumigation effects in” to “the effects of biofumigation on”

Reply: Done.

Line 289: Add the period symbol “.” after “activity [53]”

Reply: Done.

Line 291: The results were from field plot experiments? Details are needed to clarify the rate of biofumigation using cover crop, soil moisture, temperature, tarping, and treatment time.

Reply: The available information has been added and it now reads: “In a recent study, apple replant disease incidence declined in soils biofumigated with Raphanus sativus or B. juncea cover crops for two years in a site depending manner (field plot experiment, biofumigation twice a year at full bloom into moist soil, mechanical cutting and chopping of plants with a flail mulcher, immediate incorporation with a rotary cultivator; soil layering with rolls of a sowing machine, no soil tarping):…”

Line 304: Change “Among the different Brassicaceae species tested, seed meals formulations, however performed as well as fumigation treatments” to “Appropriately implemented biofumigations with particularly formulated seed meals of different Brassicaceae species performed in soil disinfestation comparably to chemical fumigation treatments”

Reply: Changed as suggested.

This manuscript is a resubmission of an earlier submission. The following is a list of the peer review reports and author responses from that submission.

Round 1

Reviewer 1 Report

The manuscript attempts to review the research advances in developing biofumigation for controlling replant disease. The topic fits the scope of Agronomy. Publication of the manuscript would draw interest and attention from the journal readers. The manuscript, however, does not focus on its major objective to review research findings on developing optimal biofumigation programs for effective replant disease control. The review is superficial, possessing few critical comments. The content does not support the conclusions. The following are suggestions for the authors to consider:

The value of this mini review would be notably enhanced with the primary goal to identify optimal biofumigation programs for effectively controlling replant disease. The manuscript should focus on reviewing research studies investigating the efficacy of biofumigation as influenced by various factors such as Brassicaceae material type, dosage, pretreatment, timing, soil properties, moisture, and temperature. Experimental designs should be briefly introduced when literature research studies are discussed. Negative results of biofumigation in the literature research should be explained with possible reasons. The growth of plants is influenced by numerous environmental factors including replant disease. How biofumigation alters soil properties and subsequently impacts plant growth should be discussed. The conclusion of this mini review is not adequately supported by its current main content. There is substantial space for improving the English writing.

Additional comments can be found in the edited version of the manuscript.

Author Response

We thank the reviewer for his/her critical comments and his/her suggestions, which very much helped us to improve the quality and focus of our review. We rewrote the manuscript based on these comments and now focus on the optimization of isothiocyanate release (Chapter 2, biofumigation) and on the optimization of biofumigation in the fight of replant disease (Chapter 4). Experimental designs are now briefly introduced when literature research studies are discussed, and for negative results now a reason is given.

The comments from the edited version of the manuscript made by the reviewer were mainly incorporated into the manuscript file and additional responses can be found in the pdf “edited version of the manuscript”.

In order to polish the English language, the manuscript was corrected by a native speaker.

Reviewer 2 Report

Please, to see the enclosed revision report file.

Author Response

GENERAL COMMENTS

This paper contains a breadth of interesting information and also creative visions relative to biofumigation by the incorporation of Brassicaceae plant materials (overall seed meal or seed pellet) into the soil is a promising preventive measure for controlling soil decline caused by replant disease. This paper also indirectly stimulates the topic of the biofumigant production from a biofuel chain based on biodiesel production from green source. I believe that a very significant challenge exists relatively to this manuscripts as regards to a circular economy system.

Reply: We are very grateful for your time and valuable comments, which we tried to implement as explained in italics below.

SPECIFIC COMMENTS

-Title. I suggest to consider this work ‘A review’ rather than ‘A mini-review’.

Reply: The title was changed as suggested.

-Keywords. Correct ‘microbiom’ in ‘microbiome’ and avoid to repeat the same words of the title.

Reply: Done.

-Paragraph 1 (The Replant Disease Syndrome). It is well structured and clearly reviewed by describing the main problem of soil decline due to replant disease. The concepts in the last sentences (Ln. 74-76) should be however emphasized and strengthen to render the manuscript more robust and well addressed to the novelty purpose of the review.

Reply: Done. We have rewritten these lines and had also to react to reviewer 1 here.

-Paragraph 2 (Biofumigation). This section of the MS seems to be quite weak under the new light of a circular economy system related to industrial production of Brassicaceae seed meal-based soil biofumigant coming from the biodiesel chain, rather than be much robust for the good information as already mentioned in the paper. As regards to improve this section, I suggest to reading the following two review articles: 1) http://dx.doi.org/10.1016/j.cropro.2014.10.025 (Par. 2.1., Fig.6) and 2) https://doi.org/10.1016/j.rser.2018.02.041 (Table 4, Fig.6, Par. 6.2.).

Reply: Thank you for providing this important and interesting further aspect. We have included an additional paragraph addressing this issue and cited the two reviews (lines 194ff. of the manuscript with changes highlighted).

-Paragraph 3 (Effects of Biofumigation on the Soil Microbiota). This section of the MS should be strongly improved after reading the Chapter 20 (pp. 413–436) of the Book published by SpringerNature In: Meghvansi MK, Varma A (Eds.) – Organic Amendments and Soil Suppressiveness in Plant Disease Management 2015, Springer Series Soil Biology, Vol. 46, ISSN 1613-3382, ISSN 2196-4831 (electronic), ISBN 978-3-319-23074-0, ISBN 978-3-319-23075-7 (eBook). Springer International Publishing Switzerland (http://doi.org/10.1007/978-3-319-23075-7) and the review article http://dx.doi.org/10.1016/j.cropro.2014.10.025 (Par. 4.1.2., 4.2.1.). In addition, in order to evaluate the effects of biofumigation on the soil microbiota, I suggest to search recent works (if exist, obviously) about the next-generation sequencing approaches by amplicon sequencing and shotgun metagenomics for assessing the soil microbiomes disturbance after biofumigation.

Reply: We are thankful for your comments. We read the suggested literature and revised the chapter after. We think, that thanks to your help the chapter significantly improved (See lines 247-254 and L.268-294 of the manuscript with changes highlighted).

Indeed amplicon sequencing and next generation sequencing has already been used to study the changes in the microbial communities after biofumigation (see Siebers et al 2018 and Yim et al 2017). We included one of these studies in the Paragraph 3 (Siebers 2018) and the other (Yim 2017) in the context of apple replant disease. We now highlighted the use of these methods in the manuscript (see lines 234 and 348 of the manuscript with changes highlighted).

-Paragraph 4 (Efficacy of Biofumigation in the Treatment of Replant Disease). Ln. 209-214. Again, the authors should provide major information on the key role of soil organic matter in determining the efficacy of biofumigation by Brassicaceae seed meals by reading the manuscript http://dx.doi.org/10.1016/j.cropro.2014.10.025 (Par. 4.2.1.).

Reply: The role of organic matter in the efficacy of biofumigation is now discussed in more detail in paragraph 2, lines 165ff of the manuscript with changes highlighted.

Ln 217. Pratylenchus penetrans? But it is a nematode, not a oomycete (Pythium) or fungus (Rhizoctonia and Cylindrocarpon) that instead are the true causal agents of replant disease!! For nematode, the authors should provide a separated section in the same paper and tune it with the title that instead talk of ‘Fighting Replant Disease’ only. Alternatively, problems caused by nematodes should be withdrawn from the text and analyzed in different paper.

Reply: Thank you for detecting the error, we corrected it (line 334 of the manuscript with changes highlighted) However, we decided to keep the nematode effects within the replant disease chapter. Nematodes were discussed as one part of the disease complex, the authors of the cited studies have analysed the effects of biofumigation on Pratylenchus penetrans and therefore, we would like to leave this part within the chapter. We have, however, corrected the misleading statement in the former line 217 of the first version of our manuscript.

-Paragraph 5 (Conclusion). I would suggest to strengthen the conclusion part so as to support strongly the future directions; otherwise, this manuscript is overall the compilation of literature reports and not benefit the readers.

Reply: Thank you, we hope that the revised conclusion section does better meet your expectations.